# Assessing Changes in Motor Function and Mobility in Individuals with Parkinson’s Disease After 12 Sessions of Patient-Specific Adaptive Dynamic Cycling

**DOI:** 10.3390/s24227364

**Published:** 2024-11-19

**Authors:** Younguk Kim, Brittany E. Smith, Lara Shigo, Aasef G. Shaikh, Kenneth A. Loparo, Angela L. Ridgel

**Affiliations:** 1Exercise Science and Exercise Physiology Program, Kent State University, Kent, OH 44242, USA; ykim23@uab.edu (Y.K.); bsmit258@kent.edu (B.E.S.); lboman@kent.edu (L.S.); 2Department of Physical Medicine and Rehabilitation, The University of Alabama at Birmingham, Birmingham, AL 35233, USA; 3Department of Neurology, Case Western Reserve University, Cleveland, OH 44106, USA; axs848@case.edu; 4Louis Stokes Cleveland VA Medical Center, Cleveland, OH 44106, USA; 5Institute for Smart, Secure and Connected Systems, Case Western Reserve University, Cleveland, OH 44106, USA; kal4@case.edu

**Keywords:** wearable technology, neurorehabilitation, motor control, movement disorders

## Abstract

Background and Purpose: This pilot randomized controlled trial evaluated the effects of 12 sessions of patient-specific adaptive dynamic cycling (PSADC) versus non-adaptive cycling (NA) on motor function and mobility in individuals with Parkinson’s disease (PD), using inertial measurement unit (IMU) sensors for objective assessment. Methods: Twenty-three participants with PD (13 in the PSADC group and 10 in the NA group) completed the study over a 4-week period. Motor function was measured using the Kinesia™ sensors and the MDS-UPDRS Motor III, while mobility was assessed with the TUG test using OPAL IMU sensors. Results: The PSADC group showed significant improvements in MDS-UPDRS Motor III scores (*t* = 5.165, *p* < 0.001) and dopamine-sensitive symptoms (*t* = 4.629, *p* = 0.001), whereas the NA group did not improve. Both groups showed non-significant improvements in TUG time. IMU sensors provided continuous, quantitative, and unbiased measurements of motor function and mobility, offering a more precise and objective tracking of improvements over time. Conclusions: PSADC demonstrated enhanced treatment effects on PD motor function compared to NA while also reducing variability in individual responses. The integration of IMU sensors was essential for precise monitoring, supporting the potential of a data-driven, individualized exercise approach to optimize treatment outcomes for individuals with PD.

## 1. Introduction

Parkinson’s disease (PD) is the second most common age-related neurological disorder [1] and is expected to double in prevalence over the next generation [2]. The primary motor symptoms of PD include bradykinesia, resting tremor, and rigidity [3]. Pharmacological and surgical therapies are effective ways to manage these symptoms. However, higher dosages of levodopa can cause dyskinesia, akinesia, confusion, hallucination, and psychosis [4,5,6,7], and deep brain stimulation carries the risks of infection, hardware failure, and an increased incidence of depression and confusion [8,9,10].

High-cadence (forced) cycling has beneficial effects on PD motor symptoms similar to pharmacological and surgical management [11,12,13,14]. In addition, it has a low risk of falling compared to exercise that requires standing. The beneficial effects of cycling on PD motor symptoms are greater when the pedaling rate (cadence) is higher than the self-selected pedaling cadence [12]. Eight weeks of forced cycling (high-cadence tandem cycling) improved UPDRS motor III score (35%), rigidity (41%), tremor (38%), and bradykinesia (28%). Similarly, high-cadence dynamic cycling (motorized stationary cycling with advanced control of cadence and power) also had significant treatment effects on PD motor symptoms, non-motor symptoms, and functional mobility [15,16].

Even though high-cadence cycling offers significant benefits for PD symptoms, it has some limitations. Forced cycling using a tandem bike is not always feasible in clinical settings due to the need for a large space and a relatively fit trainer [12], and previous studies of dynamic cycling did not adapt settings between or within exercise sessions [17]. In the current literature, periods of dynamic cycling intervention were short, usually limited to three or six sessions, which is insufficient to evaluate longer-term benefits [15,16]. Most importantly, neither of these paradigms allowed for adaptive exercise for individuals with PD. Every participant in previous studies received uniform exercise prescriptions, regardless of their physical functioning level or degree of PD motor symptoms, which led to some participants experiencing greater exercise benefits than others.

To move toward maximizing the effects and minimizing heterogeneity in individual responses, an individually tailored exercise rehabilitation model was developed [18] using two measured variables for feedback: sample entropy of cadence (SampEn) and effort (percentage of positive power). Previous findings from our lab [19,20] indicated significant associations between SampEn of cadence and motor function improvement and between the percentage of positive power and motor skill performance following the dynamic cycling intervention. A higher level of SampEn of cadence led to more positive effects on motor symptoms, and a greater percentage of power output during the dynamic cycling session improved motor function skills in individuals with PD. However, these qualitative observations do not translate directly to an individualized strategy that improves motor symptoms and function skills.

These preliminary results suggest that by quantifying the variability of cadence and identifying effort variables, PD motor function improvement can be predicted. Our previous research [21] demonstrated a significant association between high SampEn of cadence and PD motor function improvement (R^2^ = −0.545), indicating a moderately strong inverse relationship. A linear regression model further identified SampEn of cadence as the strongest and most significant predictor of PD motor function improvement in individuals with PD. These findings suggest that by calculating SampEn of Cadence during high cadence dynamic cycling sessions, we may be able to anticipate improvements in PD motor symptoms. In addition, we developed and tested an algorithm that uses SampEn of cadence and a measure of effort to determine the resistance settings for high-cadence cycling that is expected to provide benefits to the patient, referred to as patient-specific adaptive dynamic cycling (PSADC) [21].

In addition, to improve the accuracy of motor function and mobility assessments and enhance the efficacy of the PSADC paradigm, this research incorporates the use of inertial measurement unit (IMU) sensors. IMU sensors offer a highly accurate, reliable, and valid method [22] for capturing movement patterns and measuring PD motor function and mobility, as demonstrated by numerous studies showing strong reliability, validity, and discriminative ability [23,24,25]. Furthermore, it provides real-time, precise data on PD-related motor functions such as movement amplitude, rhythm and speed. These objective measurements allow us to adjust resistance levels during or after cycling sessions to achieve optimal performance for each participant. As a result, the PSADC paradigm with IMU sensors dynamically adapts to individual needs, maximizing the therapeutic effects of exercise on PD motor symptoms while also reducing variability in treatment responses across individuals.

This study is a pilot randomized controlled trial (RCT) designed to examine the effects of the PSADC paradigm on PD motor function and functional mobility in comparison to non-adaptive (NA) dynamic cycling. In addition, as a pilot study, our objective is to evaluate the feasibility and efficacy of the PSADC protocol in preparation for a larger, double-blind, randomized controlled trial (RCT). Through this process, we aim to gather critical data on the PSADC’s practicality, optimize our study design, and estimate the appropriate sample size for future research. Finally, we hypothesized that the PSADC paradigm, coupled with the objective data collected through IMU sensors, would improve the therapeutic effects of high-cadence dynamic cycling for each individual as well as reduce heterogeneity in individual changes in PD motor function and functional mobility measures.

## 2. Materials and Methods

### 2.1. Research Design

This study was a pilot randomized control trial (RCT) with two arms: (a) the PSADC group and (b) the NA group. Neither the participants nor the blinded evaluator was aware of the group allocation. It was registered with ClinicalTrials.gov (NCT05361200) and was approved by the Institutional Review Board (IRB) at Kent State University (IRB#87). Written informed consent was provided by the participants.

### 2.2. Participants

#### 2.2.1. Inclusion and Exclusion Criteria

Inclusion criteria included a diagnosis of idiopathic PD according to the UK Brain Bank Criteria [26], the ability to give written informed consent, being 50–79 years old, and a stable medical regimen of antiparkinsonian medication for at least six months. Exclusion criteria included any sign or symptoms of cardiovascular, metabolic, and/or renal disease without medical clearance from a physician.

#### 2.2.2. Sample Size Calculation

We performed an a priori power analysis using G*Power 3.1 to determine the sample size required to detect a significant difference between the two groups (PSADC and NA) on the primary outcome measure, the MDS-UPDRS Motor III score. Based on previous research [16], we used mean and standard deviation values of 14.1 ± 2.1 and 11.6 ± 1.8 for two groups as estimates of the expected effect size. We set the alpha level at 0.05 and aimed for a power of 0.8. We calculated that a sample size of at least 20 participants (10 per group) would be required to detect a statistically significant difference between the two independent means in MDS-UPDRS Motor III scores. This sample size was considered sufficient to detect a moderate effect size, ensuring reliable estimates of the intervention effect in this study.

### 2.3. Group Allocation and Exercise Intervention

Using the REDcap randomization module, we randomly assigned participants to either the PSADC or NA group based on their Hoehn and Yahr stages. Both groups completed 12 dynamic cycling sessions three times a week for four weeks. Each session included 5 min of warm-up at 60 revolutions per minute (RPM), 30 min of dynamic cycling at 80 RPM, and 5 min of cool-down at 60 RPM. The PSADC group adjusted specific resistance settings on the 3rd, 6th, and 9th sessions based on effort and SampEn of cadence variables from their previous three sessions. If the average SampEn of cadence from the previous 3 sessions was within the interquartile range of healthy adults and effort was over 65%, the resistance level increased for the next 3 sessions. If SampEn was within range but effort was below 65%, or if SampEn was below range but effort was over 65%, the same resistance level was used. When both SampEn and effort were below these thresholds, resistance was reduced for the next 3 sessions. Meanwhile, the NA group maintained the same resistance setting throughout all 12 sessions. The details of the PSADC paradigm are outlined in a study by Kim et al. [21]. Participants were blinded to the assigned group (Figure 1).

### 2.4. Outcome Measurements

#### 2.4.1. PD Motor Functions

The primary outcome measure was the Movement Disorder Society—Sponsored Revision of the Unified Parkinson’s Disease Rating Scale Motor III (MDS-UPDRS Motor III). We used this measure at baseline and after the last exercise session. To maintain objectivity, assessments were video-recorded and then evaluated by a blinded assessor who was certified in scoring the MDS-UPDRS. Video-recorded administration of the MDS-UPDRS Motor III not only upholds validity and reliability but is also cost-efficient and practical [27,28]. UPDRS Motor III rigidity measures were assessed in person by a trained evaluator. To investigate the effect of PSADC on dopamine-sensitive motor symptoms, the MDS-UPDRS Motor III test was separated into two categories of symptoms: dopamine-sensitive symptoms (such as bradykinesia, resting tremor, and rigidity, which indicate degeneration of dopaminergic neurons) and dopamine less-sensitive symptoms (such as posture, balance, and gait, which reflect the loss of nondopaminergic pathways) [29].

In addition to the clinical assessment, the Kinesia™ One device (Great Lakes Neuro Technologies, Cleveland, OH, USA) was used to provide an objective and real-time measurement of PD motor function. The Kinesia™ One system incorporates a wireless IMU sensor, which offers detailed data collection by measuring triaxial acceleration and gyroscopic movements along the X, Y, and Z axes. The IMU sensor operates at a sampling rate of 128 Hz, which allows for the capture of 128 data points per second. This is sufficient to detect both slow and rapid motor movements, which are common in individuals with PD. During the tests, the IMU sensor gathers raw data on linear acceleration and angular velocity in all three dimensions (X, Y, and Z), providing a detailed picture of the participant’s movement patterns. These raw data are continuously transmitted to a secure portal where it is processed and analyzed. The IMU sensor data evaluates motor functions like finger tapping, hand movements, and pronation–supination movements of the hands. Movement speed, amplitude, and rhythm data were collected before and after each training session, but due to session-by-session fluctuation, only baseline and post-intervention data were used for data analysis to more accurately reflect overall changes in motor function. The test results are scored on a 0–4 scale, similar to the MDS-UPDRS Motor III [30]. The maximum total score for movement speed, amplitude, and rhythm is 12 points each. A decrease in score indicates an improvement in symptom severity, whereas an increase in score indicates a worsening of symptoms.

#### 2.4.2. Mobility

Mobility was assessed with the Timed Up and Go (TUG) test combined with the OPALS system [31] (ADPM Wearable Technologies, Inc., Portland, OR), an advanced IMU sensor developed for gait analysis. The OPAL IMU sensors feature 3-axis accelerometers, gyroscopes, and magnetometers that enable precise motion detection in all three dimensions (X, Y, and Z axes). The sensors operate at a sampling rate of 128 Hz, providing high-resolution data that enables accurate monitoring of mobility and gait dynamics, such as postural changes or turning dynamics [32]. The OPAL IMU sensors were attached to the participant’s legs, arms, and trunk to track the total time of the test and to measure postural transitions, including turn duration and turn velocity. TUG was measured at baseline and immediately after the last exercise session.

#### 2.4.3. Exercise Variables

During each cycling session, heart rate (HR) was measured every two minutes using an HR monitor (Mi Band 6, Xiaomi, Beijing, China). Ratings of perceived exertion (RPE) were recorded every four minutes using a 6–20 Borg RPE scale [33]. Revolutions per minute (RPM, cadence) and power were recorded every second by a micro controller on the dynamic cycle. After each session, these values were averaged across the 30-min main set. Effort was calculated as the percentage of time a rider produced positive power to maintain the 80 RPM cadence during the session. During cycling, an animation of a balloon over water was provided immediate biofeedback to participants indicating current effort level. Participants were instructed to “keep the balloon over the water” while cycling to encourage positive power and, thus, high effort.

### 2.5. Statistical Analysis

All statistical analyses were completed using IBM SPSS version 28 (Armonk, NY, USA: IBM Corp), with an alpha level of 0.05. Independent samples *t*-tests were used to compare demographic, cycling, and physiological variables, as well as each group’s pre- and post-MDS-UPDRS Motor III change scores, to determine any differences between the two groups. For the MDS-UPDRS Motor III, dopamine-sensitive and less-sensitive PD symptoms, Kinesia motor function test, and TUG, a two-way repeated-measures analysis of variance (ANOVA; 2 groups by 2 time points) was used.

## 3. Results

Twenty-three participants completed the study. Thirteen participants completed 12 sessions of the PSADC protocol, and ten participants completed 12 sessions of the NA protocol. An independent samples *t*-test revealed no significant demographic differences between the two groups. No significant difference in the physiological and cycling variables emerged between the two groups except for HR (*t* = 2.237, *p* = 0.036) and effort (*t* = 2.113, *p* = 0.047) (Table 1).

### 3.1. MDS-UPDRS Motor III Score Changes

MDS-UPDRS Motor III scores showed a significant group by time interaction (*F* = 18.746, *df* = 1, *p* < 0.001, *η_p_*^2^ = 0.472) but no significant main effect of time (*F* = 254, *df* = 1, *p* = 0.619, *η_p_*^2^ = 0.012; Figure 2A). The PSADC group showed a 16.2% improvement (−5.3 ± 3.7 points), while the NA group showed a 14.9% decline (4.2 ± 6.7 points). This change was statistically significant (Figure 2B, *t* = −4.330, *df* = 21, *p* < 0.001). Individually, 84.6% (11/13) of the PSADC participants improved their MDS-UPDRS Motor III score, while 30% (3/10) of the NA participants improved (Figure 2C).

### 3.2. Dopamine-Sensitive and Less-Sensitive MDS-UPDRS Motor III Score Changes

Dopamine-sensitive MDS-UPDRS Motor III symptoms showed a significant group by time interaction (*F* = 14.80, *df* = 1, *p* = 0.001, *η_p_*^2^ = 0.413) but no significant main effects of time (*F* = 1.015, *df* = 1, *p* = 0.325, *η_p_*^2^ = 0.046). The PSADC group showed a 22% improvement, while the NA group showed a 14.7% decline (Figure 3A). Dopamine less-sensitive symptoms revealed significant group by time interaction (*F* = 5.097, *df* = 1, *p* = 0.035, *η_p_*^2^ = 0.195) but no significant main effects of time (*F* = 0.627, *df* = 1, *p* = 0.437, *η_p_*^2^ = 0.029). The PSADC group showed a 7.1% improvement, while the NA group showed a 20% decline (Figure 3B).

### 3.3. Kinesia One Motor Function Test Score Changes

The Kinesia One motor function test assessed the speed, amplitude, and rhythm of the movement during finger tapping, hand movement, and pronation–supination movements of hands. One participant in the PSADC group was excluded from data analysis because they had limited upper extremity range of motion and were not able to complete the test. There was no significant group by time interaction (*F* = 3.093, *df* = 1, *p* = 0.094, *η_p_*^2^ =0.134) or main effect of time (*F* = 0.272, *df* = 1, *p* = 0.607, *η_p_*^2^ = 0.013) in movement speed between the baseline (pre) and after the intervention (post). However, the PSADC group showed a 9.0% improvement, while the NA group showed a 5.0% worsening (Figure 4A). For rhythm of movement, there was no significant group by time interaction (*F* = 0.127, *df* = 1, *p* = 0.725, *η_p_*^2^ = 0.006) nor main effect of time (*F* = 3.923, *df* = 1, *p* = 0.062, *η_p_*^2^ = 0.164), but the PSADC group showed an improvement of 15.7%, and the NA group showed an improvement of 10.3% (Figure 4B). Lastly, for movement amplitude, there was a significant main effect of time (*F* = 8.452, *df* = 1, *p* = 0.009, *η_p_*^2^ = 0.297), but no significant group by time interaction (*F* = 1.852, *df* = 1, *p* = 0.189, *η_p_*^2^ = 0.085; Figure 4C).

### 3.4. Functional Mobility Changes

Functional mobility was assessed with Timed Up and Go (TUG). One participant in the PSADC group was excluded from the data analysis due to an inability to walk without assistance. TUG duration showed a significant main effect of time (*F* = 6.00, *df* = 1, *p* = 0.024, *η_p_*^2^ = 0.231) but no significant group by time interaction effect (*F* = 0.153, *df* = 1, *p* = 0.700, *η_p_*^2^ = 0.008). The PSADC group showed an 8.7% improvement, and the NA group showed a 9.0% improvement (Figure 5A). There was no significant group by time interaction (*F* = 0.010, *df* = 1, *p* = 0.922) or main effect of time (*F* = 0.109, *df* = 1, *p* = 0.745) in turn angle. There was no significant group by time interaction (*F* = 0.036, *df* = 1, *p* = 0.851, *η_p_*^2^ = 0.002) or main effect of time (*F* = 3.068, *df* = 1, *p* = 0.095, *η_p_*^2^ = 0.133) for turn duration. The PSADC groups showed a 5.2% improvement, and the NA group showed a 5.4% improvement. Similarly, for turn velocity, there was no significant group by time interaction (*F* = 0.2.792, *df* = 1, *p* = 0.110, *η_p_*^2^ = 0.123) nor the main effect of time (*F* = 0.01, *df* = 1, *p* = 0.974, *η_p_*^2^ = 0.001). However, the PSADC group showed a 9.7% improvement, and the NA group showed a 10.7% decline (Figure 5B).

## 4. Discussion

This study incorporated the use of IMU sensors alongside traditional clinical assessments of motor function and mobility in individuals with PD. This dual approach not only increases the accuracy and depth of motor function and mobility assessment but also allows for a more accurate understanding of how PD motor function responds to the PSADC paradigm over time, leading to individualized rehabilitation strategies. By combining clinical assessments with objective, quantitative data from IMU sensors, this study achieves a more comprehensive and reliable assessment of motor function in individuals with PD. In assessing mobility, the use of IMU sensors goes beyond traditional stopwatch-based assessments to provide quantitative, objective data that provides more accurate insights into how mobility is affected by PD and how it improves in response to interventions such as PSADC.

In clinical measurement, the PSADC group showed a 5.3-point improvement in the MDS-UPDRS Motor III score. This change falls within the range of a moderate clinically important difference (MCID) of 4.5–6.7 points [34]. While consistent with previous findings after three and six sessions of dynamic cycling [15,16], twelve sessions of PSADC over 4 weeks resulted in an improvement that was more than two times greater than previous findings. Moreover, 11 of 13 participants in the PSADC group experienced motor function improvement, and 9 showed an improvement within MCID. In contrast, only 3 of 10 participants in the NA group showed an improvement in their MDS-UPDRS Motor III score. The exact physiological mechanisms behind the improvements observed with PSADC remain unclear, but several hypotheses have been proposed. PSADC utilizes specific, individualized settings for high-cadence dynamic cycling based on participants’ performance, which increases the level of sample entropy (SampEn) of cadence. Previous research has demonstrated significant correlations between higher SampEn and motor function improvement [21]. By providing tailored resistance settings of high-cadence dynamic cycling, participants are exposed to a broader range of movement patterns. These varied movement patterns likely generate more complex and diverse sensory and peripheral afferent input [35,36], which is critical for neural plasticity. This increased input may lead to greater activation of the basal ganglia circuits, which play a crucial role in motor control and learning [37]. Enhanced stimulation of these circuits could facilitate better integration of motor commands and sensory feedback, resulting in improved motor performance in individuals with PD [38]. In addition, our ongoing pilot research in the lab using functional near-infrared spectroscopy (fNIRS) has demonstrated notable changes in oxyhemoglobin levels in the left prefrontal cortex following high-cadence dynamic cycling. These changes indicate alternations in cerebral blood flow, which may serve as a proxy for neuroplastic changes. Based on these preliminary findings, we postulate that the individualized resistance settings in PSADC may enhance motor function by promoting greater cortical engagement and neural plasticity, driven by increased afferent feedback and sensory stimulation.

The 12 sessions of the PSADC paradigm led to a 22% improvement in dopamine-sensitive PD MDS-UPDRS Motor III symptoms such as tremors, bradykinesia, and rigidity. This suggests that the PSADC paradigm may enhance neuroplasticity and facilitate the upregulation of dopamine production and release, like the effects of levodopa therapy, but through natural, exercise-induced pathways. This idea is consistent with previous research indicating that exercise has an effect similar to levodopa and could potentially enhance dopaminergic function in the basal ganglia of individuals with PD [39,40,41]. We observed notable improvements in movement speed and rhythm following the PSADC paradigm, consistent with previous research underscoring the beneficial effects of dynamic cycling on bradykinesia and motor timing in the PD population [15,16]. In addition, an interesting observation was the decrease in movement amplitude concurrent with the increase in speed. This phenomenon can be interpreted as participants reducing their range of motion as a compensatory mechanism to increase task performance speed. It suggests a potential trade-off between speed and amplitude, reflecting an adaptive strategy by individuals with PD to maintain or improve movement efficiency.

Notably, the NA group demonstrated a slight worsening of symptoms despite initial motor function improvement through session 6. This finding could be due to insufficient exercise intensity and lack of motivation. The non-adaptive dynamic cycling intervention might not be variable enough to elicit the physiological adaptations that we see with the PSADC paradigm. The resistance setting for the NA group was set at 1 on a scale of 1–6, and this resistance level might have been high enough to provide physiological benefits for the first six sessions [16], but participants in the NA group might also have acclimated to this low-intensity exercise, limiting the stimuli sufficient to realize the physiological benefits of high-cadence cycling. Moreover, we observed a gradual decrease in effort in some NA group participants during the second half of the intervention. This phenomenon did not emerge in the participants of the PSADC group. As a result, a significant difference in effort emerged between the two groups. We postulate that reduced effort levels and lack of variability in the resistance settings over the intervention may have limited the effect of the dynamic cycling intervention on PD motor symptoms [20]. Without sufficient variability or challenge, the NA intervention may not fully engage motor circuits or activate proprioceptive feedback mechanisms—essential for motor function improvement. In future studies, NA interventions could be enhanced by incorporating periodic adjustments to resistance levels, even if not individualized, to introduce more challenge and variability.

### 4.1. Effects of PSADC on Functional Mobility

Both the PSADC and the NA group showed improvement in TUG time to completion. These results align with previous findings [16], substantiating the positive impact of dynamic cycling on functional mobility. Although the exact physiological and biomechanical mechanisms behind mobility enhancements remain unknown, several hypotheses exist. High cadence cycling, characterized by rapid and repetitive lower body movements, is believed to augment muscle activation, sensory feedback, and motor automaticity [42,43]. These enhancements may facilitate smoother transitions in movements required for the TUG test, such as sitting to standing, walking, and turning maneuvers. Furthermore, high cadence dynamic cycling may help alleviate muscle rigidity and joint stiffness, as evidenced by improvements in the MDS-UPDRS Motor III rigidity scores [15,16]. This reduction in rigidity can enhance gait and mobility by improving walking efficiency and range of motion, which is particularly important for individuals with PD who are affected by muscle and joint rigidity. Moreover, the enhanced TUG performance observed in both groups can be attributed to the unique characteristics of the high-cadence cycling intervention. Pedaling at 80 rpm directly enhances step cadence and leg movement speed, which in turn improves spatiotemporal gait parameters such as step length, stride length, and step duration. These improvements contribute to faster gait cycles and enhance overall mobility. These findings are supported by Linder et al.’s study [44] on aerobic cycling (76 rpm), which demonstrated significant improvements in gait velocity (m/s), cadence (steps/minute), and normalized step length (cm). It is likely that high cadence dynamic cycling may also improve gait characteristics by reducing deviations and promoting more efficient gait patterns, ultimately leading to improved TUG test performance.

### 4.2. Limitations

The current study has several limitations. First, there was a large variation in cycling performance and a wide range of PD symptom severity. The severity of PD symptoms in this sample ranged from individuals with deep brain stimulation (DBS) in wheelchairs to individuals with mild PD symptoms. Although no statistical difference in demographic information emerged between the two groups, there was some variability in levodopa equivalent daily dosage (LEDD) and years with PD in the sample. These two variables often correlate with the severity of PD symptoms, so individuals with more severe motor deficits will likely show greater improvement in motor function [16]. To address this, we plan to stratify participants by PD severity (H&Y stage, baseline UPDRS scores, LEDD) and cycling performance (rider effort) to create more homogeneous groups. This approach will reduce variability in performance outcomes, allowing for a more accurate assessment of the PSADC intervention’s impact on motor function and mobility. The other limitation of this study is that all participants were on medication. We wanted to investigate the effects of exercise in real-life conditions [45], and we strictly controlled their dosage and timing for the baseline measurement, the post-exercise measurements, and the exercise session times to minimize the daily fluctuation of PD medication. In future studies, we plan to include a larger, more diverse sample, including individuals at various stages of PD and those in the “off” medication state, to assess whether the effects of the PSADC intervention extend to broader PD populations. Examining participants in the off-medication state will allow us to explore the PSADC intervention’s direct impact on motor function without the influence of pharmacological effects, providing a more comprehensive understanding of its effectiveness in PD management. The final limitation of this research is the relatively small sample size. Although the study achieved the required statistical power based on the power analysis, the results may not be fully generalizable to the broader PD population. A larger sample size would enable more definitive conclusions and a more comprehensive representation of the diverse characteristics of individuals with PD. In future research, we plan to address this limitation by recruiting a larger sample size for a double-blinded randomized controlled trial (RCT). This approach will enhance the external validity of our findings and provide more definitive evidence on the effectiveness of the PSADC intervention for improving motor function and functional mobility in individuals with PD.

## 5. Conclusions

This study is the first pilot randomized controlled study to use SampEn of cadence and effort combined with IMU sensors to evaluate the potential implications of an adaptive cycling program for individuals with PD. The incorporation of IMU sensors allowed for continuous, objective monitoring of motor function and mobility throughout the intervention, providing a more precise and detailed understanding of how participants responded to the PSADC paradigm. Not only did participants show greater improvement in PD motor function and functional mobility, but they also showed less heterogeneity in responses than the NA group. Our findings suggest that a data-driven, individualized exercise rehabilitation plan, supported by IMU sensor data, can improve and potentially maximize the effects of dynamic cycling exercise on PD motor function and mobility. For future research, we plan to conduct a larger double-blind RCT to validate these findings further. This approach will enhance the scientific validity and provide more strong evidence for the PSADC protocol. Additionally, we plan to update the PSADC algorithms using signal processing for feature extraction and machine learning (for feature selection, developing a predictive model, and optimization to maximize individualized benefits) and a larger dataset and examine the application of both within session and session to session-to-session PSADC optimizations to enhance motor function improvement. The continued integration of IMU sensor data into the PSADC paradigm will enable us to provide a more accurate, immediate, and patient-specific exercise prescription model for individuals with PD.

## 6. Patents

Angela Ridgel and Kenneth Loparo are co-inventors on two patents that are related to the device used in this study: “Bike System for Use in Rehabilitation of a Patient”, US 10,058,736.

## Figures and Tables

**Figure 1 sensors-24-07364-f001:**
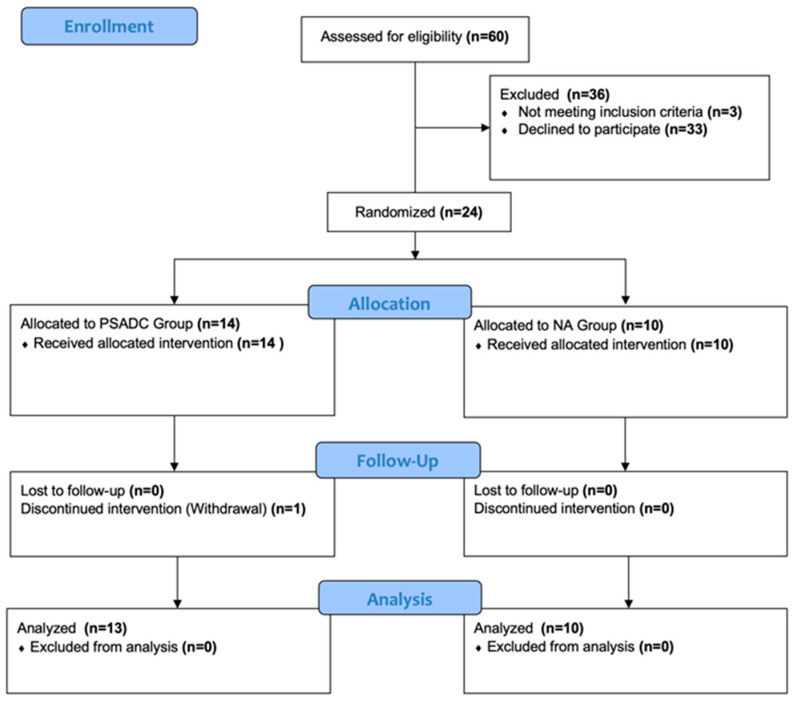
CONSORT flowchart illustrating the progression of participants through the phases of the clinical trial. Twenty-four participants were randomized into two groups: the PSADC group (*n* = 14) and the NA group (*n* = 10). All participants in the NA group completed the NA dynamic cycling intervention and the follow-up visit. In the PSADC group, 13 participants completed both the PSADC and the follow-up visit, while one participant discontinued the study.

**Figure 2 sensors-24-07364-f002:**
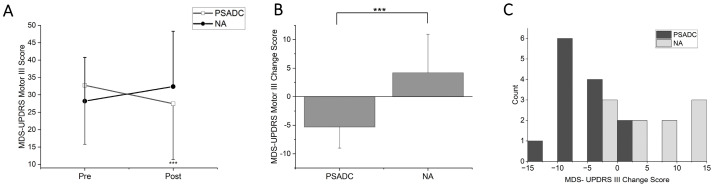
(**A**) MDS-UPDRS Motor III score changes for the PSADC (black circles) and NA (white squares) groups. (**B**) Changes between groups. The PSADC group showed improvement, as indicated by a decrease in scores, and the NA group showed a slight increase. Error bars = standard deviation. ***, *p* < 0.001. (**C**) MDS-UPDRS Motor III Score change histogram. Improvements are shown as negative values, and worsening is illustrated as positive. PSADC = black bars, NA = gray bars.

**Figure 3 sensors-24-07364-f003:**
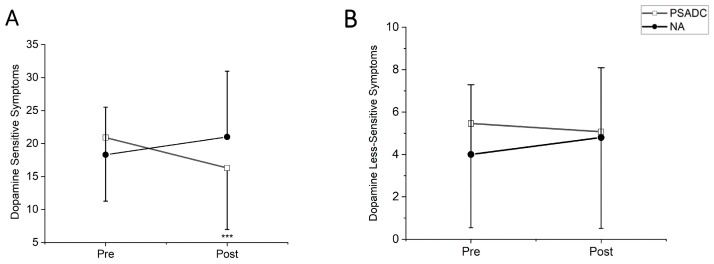
(**A**) MDS-UPDRS Motor III Dopamine Sensitive Symptom Scores. The PSADC group (black circles) exhibited a decrease in symptoms post-intervention, while the NA group (white squares) exhibited a slight increase. Error bars represent the standard deviation, highlighting the variability within each group. The decrease in the PSADC group was statistically significant (*** *p* < 0.001). (**B**) MDS-UPDRS Motor III Dopamine Less-Sensitive Symptom Scores. Both the PSADC (black circles) and NA (white squares) groups show minimal changes.

**Figure 4 sensors-24-07364-f004:**
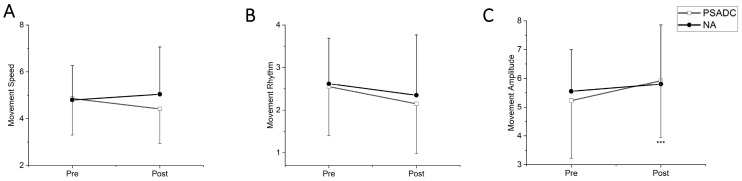
(**A**) Movement speed score: The PSADC group improved in movement speed after the intervention. Conversely, the NA group shows a worsening of symptoms. Error bars = standard deviation. (**B**) Movement rhythms score: The total score for movement speed, rhythm, and amplitude is 12 points each. A decrease in score (improvement) is observed in the PSADC group post-intervention. NA group scores were unchanged. (**C**) Movement amplitude score: Movement amplitude scores show a significant increase in the PSADC group post-intervention compared to the NA group (***, *p* < 0.001).

**Figure 5 sensors-24-07364-f005:**
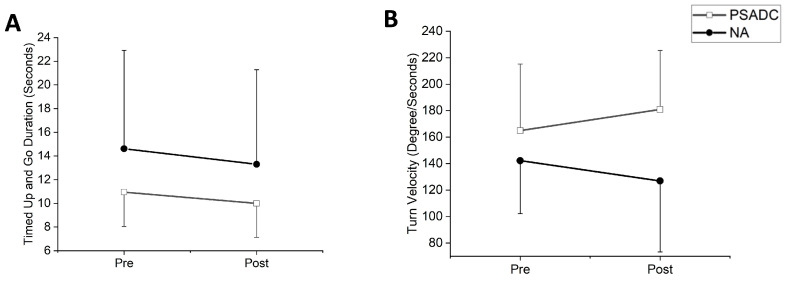
(**A**) TUG test time: The PSADC group (white squares) and the NA group (black circles) showed a reduction in time at post-intervention. (**B**) Turn velocity: Turn velocity shows an increase for the PSADC group, while the NA did not change. Error bars = standard deviation.

**Table 1 sensors-24-07364-t001:** Participants’ Demographic, PD-related, and Cycling Variables.

Variables	PSADC (*n* = 13)	NA (*n* = 10)	*p*-Value
Ages (Years)	72.08 ± 5.56	72.20 ± 4.82	0.956
Height (cm)	175.64 ± 12.48	177.42 ± 8.71	0.706
Weight (kg)	78.71 ± 25.23	88.82 ± 16.77	0.288
BMI (kg/m^2^)	25.54 ± 5.48	27.69 ± 3.52	0.297
H&Y (stage)	2.38 ± 1.12	2.10 ± 0.56	0.473
LEDD (mg)	654.00 ± 563.51	510.00 ± 338.13	0.484
Years with PD (Years)	8.95 ± 6.63	4.39 ± 3.91	0.071
Pedaling Speed (RPM)	79.42± 0.06	79.42 ± 0.09	0.813
Power (W)	21.33 ± 18.11	7.13 ± 18.40	0.078
Effort (%)	84.59 ± 30.64	54.18 ± 38.46	**0.047**
SampEn of Cadence	1.64 ± 0.14	1.57 ± 0.10	0.217
RPE	11.96 ± 1.52	12.78 ± 0.87	0.145
HR (BPM)	85.77 ± 6.54	80.23 ± 4.89	**0.036**

Values are reported as mean ± standard deviation. Bold values are *p* < 0.05. Abbreviations: PSADC, patient-specific adaptive dynamic cycling; NA, non-adaptive; H&Y, Hoehn and Yahr; BMI, Body Mass Index; LEDD, levodopa equivalent daily dose; RPM, revolutions per minute; W, Watts; SampEn, sample entropy; RPE, Rate of Perceived Exertion; HR, heart rate; BPM, Beats Per Minute.

## Data Availability

Deidentified data may be available upon request.

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
