# Peer review of "Assessing Changes in Motor Function and Mobility in Individuals with Parkinson’s Disease After 12 Sessions of Patient-Specific Adaptive Dynamic Cycling"

_sensors, 2024, doi:10.3390/s24227364_

Round 1
Reviewer 1 Report
Comments and Suggestions for Authors
This is an interesting paper. I have some issues with this paper which should be addressed before acceptance.
1. The power analysis is not clear. Which means and SD's do you mean? Which outcome? You mention: "The analysis indicated that a sample size of 20 participants would be necessary." Necessary for what exactly...? Not clear.
2. How did you handle missing data, you had one drop-out
3. Add reference for UK Brain Bank criteria
4. How did You define the ON Stage?
5. Did You include patients with any cognitive deficits (cannot see MMSE or MoCA values...
Abstract: Abbreviation IMU, please explain. What do you mean by "more accurate monitoring of improvements, provide evidence in the abstract soon already. What do you mean by feasibility of a data-driven approach, I don't see any feasibility results in the abstract.
Introduction: You mention that a higher level of cadence sampling led to more positive effects on motor symptoms. Which ones do you mean, please explain. Please explain what you mean by the fact that the improvement of motor function in PD can be predicted, and that this predictive model can then (you spell them wrong) be used to improve physiological benefits, which ones? You mention IMU in the introduction, but it would be interesting to present the ones that have been validated so far in PD? Are they really that reliable? Your hypothesis that the PSADC paradigm coupled with objective data collected via IMU would improve therapeutic effects needs more explanation. Explain how an objective measurement can lead to improved therapeutic effects. How can you also reduce the heterogeneity of individual changes? This is not clear.
Materials and methods: Who was responsible for the diagnosis of Parkinson's disease. The calculation of the sample size is not clear. Please explain why you chose to do 12 sessions. And also why four weeks seems very short. Previous studies used a much longer design and used a much larger cohort, which even led to neutral (negative) results...
Results: why 13 in one group and only 10 in the other group... Figures 4a, b, c, what exactly do the numbers on the y-axis mean? The results section does not specify exactly which data from the KINESIA were used. You mention in the methods section that before and after each training, data was used... this seems like a lot of data acquired...? I guess there were fluctuations?
Discussion: Could you explain what the real benefit of IUMs is? The idea that PSADC can improve neuroplasticity is totally hypothetical since no physiological measurements have now been made. Objective IMU measurements actually show positive effects for both trainings on TUG (8.7% vs. 9%), as well as for movement rate, both groups improved. This is not discussed. Overall, some effects (in favor of the PSADC group…) could simply be explained by the difference in effort.
Conclusions: remove the fact that your study assessed the “effectiveness and feasibility” of an adaptive cycling program… Effectiveness can only be measured by a well-powered RCT and I see also no feasibility measures in this pilot study.
Author Response
|
Reviewer 1’s Revision Request |
Response |
|
The power analysis is not clear. Which means and SD's do you mean? Which outcome? You mention: "The analysis indicated that a sample size of 20 participants would be necessary." Necessary for what exactly...? Not clear. |
We used Ridgel & Ault (2019)’s data for the power analysis. Specifically, we used MDS-UPDRS Motor 3 mean and S.D value since it is the major outcome of this research. The mean and S.D for both groups are 14.1±2.1, and 11.6 ± 1.8. Based on these values, we could calculated the sample size of 20 participants (10 group each) to detect a statistically significant difference between the two means in MDS-UPDRS Motor 3 scores. |
|
How did you handle missing data, you had one drop-out |
We encountered one participant withdrawal from the PSADC group, who was unable to complete the follow-up testing. As this participant did not provide any post-intervention data, we excluded their data entirely from the analysis. Consequently, analyses were conducted only with data from participants who completed all baseline and follow-up assessments. |
|
Add reference for UK Brain Bank criteria |
Thank you for your feedback. We have added reference for UK Brain Bank criteria. |
|
How did You define the ON Stage? |
The "ON" stage was defined as the period when participants were taking their antiparkinsonian medication. Participants were instructed to follow their regular medication schedules, and we confirmed their ON stage status during exercise session and testing session to maintain consistency in motor performance and ensure reliable data. |
|
Did You include patients with any cognitive deficits (cannot see MMSE or MoCA values...
|
We did not include individuals with PD who exhibited cognitive deficits. Although we did not administer specific cognitive assessments such as the MoCA or MMSE, all participants were screened to ensure they met cognitive criteria, including the ability to understand and independently provide informed consent. This screening confirmed that participants were free from cognitive impairments that could interfere with their comprehension of instructions or their engagement in the study protocol. |
|
Abstract: Abbreviation IMU, please explain. What do you mean by "more accurate monitoring of improvements, provide evidence in the abstract soon already. What do you mean by feasibility of a data-driven approach, I don't see any feasibility results in the abstract |
IMU Explanation and Abbreviation: Inertial Measurement Unit (IMU) sensor is defined early on to ensure clarity on its purpose and use. More Accurate Monitoring of Improvements” Clarified: We have replaced the ‘accurate’ to ‘precise and objective’ to clarify that the IMU sensors provided continuous, quantitative monitoring, allowing for a more detailed and unbiased assessment of improvements over time. Feasibility of Data-Driven Approach Clarified: For clarification, we have changed the word “feasibility” to “potential” to better convey that the study demonstrates early promise for a data-driven, individualized exercise approach in PD rehabilitation. |
|
Introduction: You mention that a higher level of cadence sampling led to more positive effects on motor symptoms. Which ones do you mean, please explain. Please explain what you mean by the fact that the improvement of motor function in PD can be predicted, and that this predictive model can then (you spell them wrong) be used to improve physiological benefits, which ones? You mention IMU in the introduction, but it would be interesting to present the ones that have been validated so far in PD? Are they really that reliable? Your hypothesis that the PSADC paradigm coupled with objective data collected via IMU would improve therapeutic effects needs more explanation. Explain how an objective measurement can lead to improved therapeutic effects. How can you also reduce the heterogeneity of individual changes? This is not clear |
Explanations of SampEn of cadence and PD motor function improvement.
Our previous research (Kim et al., 2024) demonstrated that a high level of sample entropy (SampEn) of cadence is significantly associated with improvements in PD motor function, with an R2=−0.545, indicating a moderately strong inverse relationship. A linear regression model further identified SampEn of cadence as the strongest and most significant predictor of motor function improvement in individuals with PD. These findings suggest that by calculating SampEn of cadence during cycling sessions, we may be able to anticipate potential improvements in motor symptoms.
Reducing Heterogeneity in Individual Responses IMU-driven data allows us to prescribe optimal resistance settings of dynamic cycling based on their unique motor patterns. This individualized adjustment helps to minimize variability in therapeutic outcomes, as participants receive an intervention that is optimally matched to their motor performance, thereby reducing variation in treatment response across the group. We added this information in introduction part.
Explanations of integrating IMU into PSADC The benefit of integrating IMU sensors into the PSADC paradigm is that it provides real-time, precise data on PD-related motor functions such as movement amplitude, rhythm and speed. These objective measurements allow us to adjust resistance levels during or after cycling sessions to achieve optimal performance for each participant. As a result, the PSADC paradigm can dynamically adapt to individual needs, maximizing the therapeutic effects of exercise on PD motor symptoms.
Validation of using IMU sensors in PD populations.
According to numerous studies, IMU sensors used to measure functional mobility and PD-related motor functions in individuals with Parkinson’s disease have shown strong reliability, validity, and discriminative ability.
Heldman, D. A., Espay, A. J., LeWitt, P. A., & Giuffrida, J. P. (2014). Clinician versus machine: reliability and responsiveness of motor endpoints in Parkinson's disease. Parkinsonism & related disorders, 20(6), 590–595. https://doi.org/10.1016/j.parkreldis.2014.02.022
Bailo, G.; Saibene, F.L.; Bandini, V.; Arcuri, P.; Salvatore, A.; Meloni, M.; Castagna, A.; Navarro, J.; Lencioni, T.; Ferrarin, M.; et al. Characterization of Walking in Mild Parkinson’s Disease: Reliability, Validity and Discriminant Ability of the Six-Minute Walk Test Instrumented with a Single Inertial Sensor. Sensors 2024, 24, 662. https://doi.org/10.3390/s24020662
Mostile, G., Giuffrida, J. P., Adam, O. R., Davidson, A., & Jankovic, J. (2010). Correlation between Kinesia system assessments and clinical tremor scores in patients with essential tremor. Movement disorders : official journal of the Movement Disorder Society, 25(12), 1938–1943. https://doi.org/10.1002/mds.23201
|
|
Materials and methods: Who was responsible for the diagnosis of Parkinson's disease. The calculation of the sample size is not clear. Please explain why you chose to do 12 sessions. And also why four weeks seems very short. Previous studies used a much longer design and used a much larger cohort, which even led to neutral (negative) results...
|
Diagnosis of Parkinson’s Disease All participants had a confirmed diagnosis of PD made by a licensed physician prior to enrollment in the study. At baseline, we reviewed each participant’s medication list to verify their levodopa equivalent daily dose (LEDD) and current medication status, ensuring consistency across participants regarding their PD management.
Sample Size Calculation As addressed above, our sample size calculation was based on previous research, with power analysis conducted to ensure adequate statistical power to detect meaningful differences between groups on the primary outcome measure, the MDS-UPDRS Motor III score. This analysis indicated that a sample size of 20 participants would be sufficient to capture significant effects within the scope of this pilot study.
Rationale for 12 Sessions and Four-Week Duration We acknowledge that other studies have implemented longer exercise interventions with larger cohorts, yet some have yielded neutral or mixed outcomes. For this pilot study, we chose a four-week intervention with 12 sessions of high-cadence dynamic cycling to focus on the relatively acute effects of this intervention on PD motor function and functional mobility. Previous studies on high-cadence dynamic cycling have typically been limited to even shorter periods, often only two weeks. By extending to four weeks, we aimed to explore both immediate and short-term changes, establishing a foundational understanding of PSADC’s effects.
In future research, we plan to extend the intervention duration to assess potential long-term effects. High-cadence dynamic cycling incorporates velocity-based motor training at increased cadence, which is particularly beneficial for individuals with neurological conditions. This approach aims to promote neuroplasticity and improve motor outcomes in PD more effectively than traditional exercise interventions, justifying our focus on a shorter yet intensive regimen for this pilot study.
|
|
Results: why 13 in one group and only 10 in the other group... Figures 4a, b, c, what exactly do the numbers on the y-axis mean? The results section does not specify exactly which data from the KINESIA were used. You mention in the methods section that before and after each training, data was used... this seems like a lot of data acquired...? I guess there were fluctuations?
|
Group Allocation and Participant Numbers We used the REDCap randomization tool to assign participants to the two groups, with randomization stratified by each participant’s Hoehn and Yahr (H&Y) stage to ensure balanced PD motor symptom severity across groups. By accounting for H&Y stages, we were able to minimize differences in PD-related characteristics, such as H&Y stage and levodopa equivalent daily dose (LEDD), resulting in no significant differences between the groups on these variables. Although this process led to a slight imbalance in participant numbers (13 in one group and 10 in the other), we ensured that each group had at least 10 participants to maintain the statistical power established in our a priori power analysis. This design allowed us to conduct a robust comparison of the intervention effects between the two groups.
Explanation of Y-Axis in Figure 4 (Panels A, B, C) In Figure 4, the y-axis represents the total scores for movement speed, amplitude, and rhythm, measured by the Kinesia™ system. Higher scores indicate greater symptom severity, with lower scores reflecting better motor performance. The maximum total score for each of these measures is 12 points. Therefore, a decrease in score post-intervention reflects an improvement in symptoms. We have explained detailed information regarding the total score of kinesia scores.
Explanations of Kinesia Dataset Thank you for highlighting this point. We collected Kinesia Motor Function test data before and after each training session, resulting in a total of 24 Kinesia datasets per participant. However, due to fluctuations in measurements across sessions, this session-by-session data did not yield statistically significant or meaningful outcomes. Therefore, for data analysis, we focused on the baseline Kinesia data and the data collected after the 12th session to provide a clearer picture of overall changes in motor function. This approach has been clarified in the Methods section.
Following sentence was added for the clarification of using Kinesia data.
“Movement speed, amplitude, and rhythm data were collected before and after each training session, but due to session-by-session fluctuation, only baseline and post-intervention data were used for data analysis to more accurately reflect overall changes in motor function.” |
|
Discussion: Could you explain what the real benefit of IUMs is? The idea that PSADC can improve neuroplasticity is totally hypothetical since no physiological measurements have now been made. Objective IMU measurements actually show positive effects for both trainings on TUG (8.7% vs. 9%), as well as for movement rate, both groups improved. This is not discussed. Overall, some effects (in favor of the PSADC group…) could simply be explained by the difference in effort.
|
Physiological measurements We have added following information to further explain about physiological measurement regarding high cadence dynamic cycling and PD motor function.
Improvements in TUG for both groups
Moreover, the enhanced TUG performance observed in both groups can be attributed to the unique characteristics of the high-cadence cycling intervention. Pedaling at 80 rpm directly enhances step cadence and leg movement speed, which in turn improves spatiotemporal gait parameters such as step length, stride length, and step duration. These improvements contribute to faster gait cycles and enhanced overall mobility.
|
|
Conclusions: remove the fact that your study assessed the “effectiveness and feasibility” of an adaptive cycling program… Effectiveness can only be measured by a well-powered RCT and I see also no feasibility measures in this pilot study.
|
Thank you for your feedback. We have replaced the term "effectiveness and feasibility" in the conclusions. We recognize that effectiveness requires a well-powered randomized controlled trial (RCT) and that this pilot study did not include specific feasibility measures. The revised conclusion now emphasizes the observed improvements in motor function and the potential implications of the PSADC paradigm for future research. |
Reviewer 2 Report
Comments and Suggestions for Authors
This manuscript presents an interesting pilot randomized controlled trial on the effects of patient-specific adaptive dynamic cycling (PSADC) on motor function and mobility in patients with Parkinson’s disease (PD), with a focus on the use of inertial measurement unit (IMU) sensors. The study has several strengths, but also some areas that require further attention and clarification.
1. While the authors discuss some possible physiological mechanisms behind the observed improvements in the PSADC group, more in-depth exploration and discussion of the underlying mechanisms would enhance the manuscript. For example, how exactly does the individualized resistance setting based on SampEn of cadence and effort translate into improved motor function and neuroplasticity?
2. The discussion of the lack of improvement or even slight worsening in the NA group could be more detailed. What are the specific limitations of the non-adaptive cycling intervention in terms of physiological adaptations, and how could these be addressed in future studies?
3. The manuscript suggests that adding an IMU during data collection improves data accuracy, but there is no control experiment to demonstrate the necessity of using an IMU in data collection. We recommend conducting an experiment to show the impact of adding the IMU on the assessment scores
4. The limitations of the study are clearly stated, but some could be further elaborated. For instance, the large variation in cycling performance and PD symptom severity could have a more detailed discussion on how it might have affected the results and what steps could be taken in future research to mitigate this issue.
5. Given the relatively small sample size and the fact that all participants were on medication, the generalizability of the findings to a broader PD population should be more critically discussed.
6. Please redraw Figure 1, as the text in the figure is too small to read clearly.
Comments on the Quality of English LanguageThere are a few minor grammar and spelling errors throughout the manuscript that should be corrected. For example, 'PDADC' in line 199 should be 'PSADC'. The units for BMI in Table 1 are incorrect. References 15 and 16 should be placed before the period.
Author Response
|
Reviewer 2 Revision Request |
Response |
|
While the authors discuss some possible physiological mechanisms behind the observed improvements in the PSADC group, more in-depth exploration and discussion of the underlying mechanisms would enhance the manuscript. For example, how exactly does the individualized resistance setting based on SampEn of cadence and effort translate into improved motor function and neuroplasticity?
|
We added this information in discussion part to explain about physiological mechanisms of improvements of PD motor function in PSADC group.
“In addition, our ongoing pilot research in the lab using functional near-infrared spectroscopy (fNIRS) has demonstrated notable changes in oxyhemoglobin levels in the left prefrontal cortex following high-cadence dynamic cycling. These changes indicate alternations in cerebral blood flow, which may serve as a proxy for neuroplastic changes. Based on these preliminary findings, we postulate that the individualized resistance settings in PSADC may enhance motor function by promoting greater cortical engagement and neural plasticity, driven by increased afferent feedback and sensory stimulation.”
|
|
The discussion of the lack of improvement or even slight worsening in the NA group could be more detailed. What are the specific limitations of the non-adaptive cycling intervention in terms of physiological adaptations, and how could these be addressed in future studies? |
Thank you for your feedback. We have added following sentence to further explain about the limitation of NA intervention and future directions.
|
|
The manuscript suggests that adding an IMU during data collection improves data accuracy, but there is no control experiment to demonstrate the necessity of using an IMU in data collection. We recommend conducting an experiment to show the impact of adding the IMU on the assessment scores
|
Thank you for your suggestion. While we did not include a control experiment without IMU sensors, existing literature indicates that traditional methods for measuring the TUG test, such as using a stopwatch, can introduce potential biases, including timing inaccuracies and fails to detect subtle movement abnormalities in PD. IMUs offer continuous, objective measurement, which enhances data precision and consistency in motor assessments for individuals with PD. In future studies, we plan to incorporate a control condition without IMU data to more directly demonstrate the impact of IMU integration on assessment accuracy. Golder, J., et al. (2020). Towards Using the Instrumented Timed Up-and-Go Test for Screening of Sensory System Performance for Balance Control in Older Adults. MDPI. |
|
The limitations of the study are clearly stated, but some could be further elaborated. For instance, the large variation in cycling performance and PD symptom severity could have a more detailed discussion on how it might have affected the results and what steps could be taken in future research to mitigate this issue.
|
Thank you for pointing this out. We agree that the variation in cycling performance and PD symptom severity among participants warrants further discussion. This variability could have influenced the results by introducing a wider range of responses to the PSADC intervention. Differences in PD severity, such as motor symptom severity and their cycling performance, may have contributed to varying levels of engagement or effectiveness in the cycling sessions.
We have added following sentences. “To address this, we plan to stratify participants by PD severity (H&Y stage, baseline UPDRS scores, LEDD) and cycling performance (rider effort) to create more homogeneous groups. This approach will reduce variability in performance outcomes, allowing for a more accurate assessment of the PSADC intervention's impact on motor function and mobility.” |
|
Given the relatively small sample size and the fact that all participants were on medication, the generalizability of the findings to a broader PD population should be more critically discussed. |
Thank you for your feedback. We have added following sentences to discussion part for clarifications
“In future studies, we plan to include a larger, more diverse sample, including individuals at various stages of PD and those in the "off" medication state, to assess whether the effects of the PSADC intervention extend to broader PD populations. Examining participants in the off-medication state will allow us to explore the PSADC intervention's direct impact on motor function without the influence of pharmacological effects, providing a more comprehensive understanding of its effectiveness in PD management.” |
|
Please redraw Figure 1, as the text in the figure is too small to read clearly.
|
We have created a new consort diagram with clearer figure. |
|
Comments on the Quality of English Language There are a few minor grammar and spelling errors throughout the manuscript that should be corrected. For example, 'PDADC' in line 199 should be 'PSADC'. The units for BMI in Table 1 are incorrect. References 15 and 16 should be placed before the period.
|
Thank you for your feedback. We have corrected the spelling errors and BMI unit errors. |
Round 2
Reviewer 1 Report
Comments and Suggestions for Authors
No further comments. Well done
Reviewer 2 Report
Comments and Suggestions for Authors
In my opinion, the manuscript is acceptable in the present form.